# Quantitative Analysis of Plasma Cell-Free DNA and Its DNA Integrity and Hypomethylation Status as Biomarkers for Tumor Burden and Disease Progression in Patients with Metastatic Neuroendocrine Neoplasias

**DOI:** 10.3390/cancers14041025

**Published:** 2022-02-17

**Authors:** Esther Mettler, Christian Fottner, Neda Bakhshandeh, Anja Trenkler, Robert Kuchen, Matthias M. Weber

**Affiliations:** 1Department of Endocrinology and Metabolism, I Medical Clinic, University Hospital, Johannes Gutenberg University of Mainz, 55131 Mainz, Germany; christian.fottner@unimedizin-mainz.de (C.F.); neda.bakhshandeh@unimedizin-mainz.de (N.B.); anja.trenkler@unimedizin-mainz.de (A.T.); matthias.weber@unimedizin-mainz.de (M.M.W.); 2Institute of Medical Biostatistics, Epidemiology and Informatics, University Medical Center of the Johannes Gutenberg-University, 55131 Mainz, Germany; robert.kuchen@uni-mainz.de

**Keywords:** cell-free DNA, methylation, DNA integrity, neuroendocrine neoplasia

## Abstract

**Simple Summary:**

Neuroendocrine neoplasias (NEN) are a heterogeneous group of frequent slow-progressing malignant tumors for which a reliable marker for tumor relapse and progression is still lacking. Previously, circulating cell-free DNA and its global methylation status and fragmentation rate have been proposed to be valuable prognostic tumor markers in a variety of malignancies. In the current study, we compared plasma cell-free DNA (cfDNA) properties of NEN patients with a healthy control group and a group of surgically cured patients. Our results revealed significantly higher plasma cfDNA concentrations with increased fragmentation and hypomethylation in patients with advanced metastatic NEN, which was strongly associated with tumor load and could help to differentiate between metastasized disease and presumably cured patients. This suggests that the combined analysis of plasma cfDNA characteristics is a potent and sensitive prognostic and therapeutic biomarker for tumor burden and disease progression in patients with neuroendocrine neoplasias.

**Abstract:**

Background: Neuroendocrine neoplasia (NEN) encompasses a diverse group of malignancies marked by histological heterogeneity and highly variable clinical outcomes. Apart from Chromogranin A, specific biomarkers predicting residual tumor disease, tumor burden, and disease progression in NEN are scant. Thus, there is a strong clinical need for new and minimally invasive biomarkers that allow for an evaluation of the prognosis, clinical course, and response to treatment of NEN patients, thereby helping implement individualized treatment decisions in this heterogeneous group of patients. In the current prospective study, we evaluated the role of plasma cell-free DNA concentration and its global hypomethylation and fragmentation as possible diagnostic and prognostic biomarkers in patients with neuroendocrine neoplasias. Methods: The plasma cfDNA concentration, cfDNA *Alu* hypomethylation, and *LINE-1* cfDNA integrity were evaluated prospectively in 63 NEN patients with presumably cured or advanced metastatic disease. The cfDNA characteristics in NEN patients were compared to the results of a group of 29 healthy controls and correlated with clinical and histopathological data of the patients. Results: Patients with advanced NEN showed a significantly higher cfDNA concentration and percentage of *Alu* hypomethylation and a reduced *LINE-1* cfDNA integrity as compared to the surgically cured NET patients and the healthy control group. The increased hypomethylation and concentration of cfDNA and the reduced cfDNA integrity in NEN patients were strongly associated with tumor burden and poor prognosis, while no correlation with tumor grading, differentiation, localization, or hormonal activity could be found. Multiparametric ROC analysis of plasma cfDNA characteristics was able to distinguish NEN patients with metastatic disease from the control group and the cured NEN patients with AUC values of 0.694 and 0.908, respectively. This was significant even for the group with only a low tumor burden. Conclusions: The present study, for the first time, demonstrates that the combination of plasma cfDNA concentration, global hypomethylation, and fragment length pattern has the potential to serve as a potent and sensitive prognostic and therapeutic “liquid biopsy” biomarker for tumor burden and disease progression in patients with neuroendocrine neoplasias.

## 1. Introduction

Neuroendocrine neoplasms (NENs) represent a heterogeneous group of malignancies with various clinical presentations and prognoses. They derive from hormonal and nervous system cells and can occur in all organs. The incidence rate of NENs is approximately 2.5–5/100,000 per year [1]. Although they still fulfill the definition of a rare disease, disease incidence is continuously rising, partially due to the improved techniques for tumor detection. Approximately 65% of all NENs originate from the digestive tract because neuroendocrine cells show the highest density at these sites. Even when considered to have a favorable prognosis due to slow tumor growth, the mean 5-year survival rate is only ~67% (range 15–95%) and depends on several factors, including tumor grading, staging, site of origin, and the presence of metastases [1,2].

According to the World Health Organization (WHO, Geneva, Switzerland) 2019 classification, NENs are subdivided based on histopathological grading into well–moderately differentiated neuroendocrine tumors (NET G1-G3) and neuroendocrine carcinomas (NECs), which improves therapeutic decisions and evaluation of prognosis [3]. 

Chromogranin A (CgA) has been accepted as the most commonly used general tumor marker for the diagnosis, follow-up, and treatment monitoring of NEN patients, especially with non-functional tumors. However, its use is hampered by variable sensitivity and poor specificity. Furthermore, CgA can be elevated unspecifically in many other conditions, such as renal insufficiency, atrophic gastritis, and proton pump inhibitor therapy. In addition, its plasma concentration is modified by the antisecretory effect of somatostatin analogs, which are widely used in the treatment of differentiated NET [4].

Therefore, more sensitive and specific markers for the evaluation and therapeutic management of patients with NEN are urgently needed. In this context, “liquid biopsies”, with the analysis of tumor material obtained non- or minimally invasively from blood or other body fluids, may provide next-generation prognostic, diagnostic, and therapeutic biomarkers for the implementation of individualized precision medicine in oncology, without the need for serial tumor tissue biopsies or imaging [5,6,7]. 

Currently, the genomic analysis or the molecular characterization of specific tumor-associated genes from circulating tumor cells (CTC) or circulating tumor DNA (ctDNA) is a widely explored and promising tool towards the implementation of liquid biopsies in personalized oncology with regard to the identification of diagnostic genetic patterns or drugable targets. 

In neuroendocrine neoplasias, the vast majority of studies of liquid biopsies so far have focused on the use of circulating tumor cells (CTCs) as a surrogate parameter for tumor burden and as a source for the genomic or molecular profiling of the tumor DNA. Less frequently, circulating free tumor DNA (ctDNA) or RNA has been used for transcript profiling or genomic analysis of the tumors [8]. However, in contrast to other tumor entities like breast or lung cancer, where analysis of CTCs is already an established tool for the diagnosis and management of the patients, the potential role of liquid biopsies in the management of patients with NEN still needs to be established. The major limitations of CTC quantification and the genetic profiling of circulating tumor DNA are the low number and detection rate of TCs and the rarity of targetable alterations in NENs [8]. Therefore, taking a more general approach with the characterization of the total circulating cell-free DNA plasma concentration and its methylation and fragmentation status may offer a more feasible and comprehensive insight into a patient’s tumor disease and may even provide information about the tumor microenvironment [6].

cfDNA is released from cells mostly through apoptosis, active cellular release, autophagy, and necrosis. cfDNA can be detected at a low concentration in many body fluids, including plasma, where it has been shown to have a short half-life of less than 2.5 h. In healthy individuals, the majority of circulating plasma cfDNA is derived from hematopoietic cells, especially after intense exercise [6]. Elevated cfDNA plasma concentrations correlating with tumor burden and poor prognosis have been reported for many metastatic cancers [9,10,11]. Although the exact mechanism remains to be clarified, the high degree of necrosis in many advanced tumors and the release of tumor DNA fragments via phagocytosis of necrotic neoplastic cells might contribute to the increased release of cell-free tumor DNA into the circulation. Therefore, an increased cfDNA plasma concentration has been reported to be a promising tumor marker for non-invasive cancer diagnosis, evaluation of prognosis, and therapy monitoring. In addition, the fragmentation and methylation pattern of cfDNA derived from tumor cells seem to be different from cfDNA derived from healthy cells. DNA methylation occurs mainly on CpG motifs, where a guanine base immediately follows a cytosine base in the 3′ direction. There are over 28 million CpG dinucleotides in humans, of which up to 80% are methylated. CpG islands are restricted to only two locations in DNA: in a methylated form in the long repetitive sequences of non-coded DNA sections and also in the promoter area of a gene [12,13,14]. Most of the human genome is methylated, but in a wide range of pathologies, including cancer, a global DNA hypomethylation occurs. This hypomethylation affects, to a large extent, repetitive elements, which constitute ~45% of the genome. It was demonstrated that the methylation of different repetitive sequences, namely, *LINE-1*, *Alu*, and satellite 2 (Sat2), significantly correlated with global methylation levels [15]. Therefore, these repeats can be used as surrogate reporters of global methylation [16], and increased rates of global hypomethylation have been reported in the tissue [17,18,19,20,21] and blood [22,23,24,25] of many cancer patients, whereas little is known about global hypomethylation of cfDNA in cancer patients [26,27,28].

*Alu* elements are the most abundant repeats, with more than 1 million copies per haploid genome and spanning ~10% of the genome sequence [29]. *Alu* repeats tend to accumulate in gene-rich regions [30,31] and harbor ~25% of all CpG dinucleotides in the human genome [16]. Since it is has been shown that cfDNA from tumor cells presents with a different fragmentation profile than cfDNA from healthy cells, amplificons of repetitive DNA elements such as *Alu* or *LINE-1* have been used to analyze cfDNA integrity as a metric measure of size distribution in a variety of tumors [32,33,34,35]. Although conflicting data for the size distribution of cfDNA in cancer patients exists, it is now commonly accepted that circulating free tumor DNA exhibits a higher fragmentation than the cfDNA shed by non-neoplastic tissues [35,36,37,38,39,40]. Thus, the cfDNA integrity and the hypomethylation status of repetitive DNA sequences like *Alu* or *LINE-1* in cfDNA have recently been evaluated as an attractive non-invasive biomarker for tumor diagnosis and prognosis in various types of cancer. However, up to now, no information is available on the role of plasma cfDNA methylation and integrity as a potential liquid biopsy biomarker in patients with neuroendocrine neoplasias.

In the current study, we, therefore, evaluated the plasma cfDNA concentration, cfDNA *Alu* hypomethylation, and *LINE-1* cfDNA integrity as a diagnostic and prognostic biomarker in 63 patients with neuroendocrine neoplasias as well as in a healthy control group. When comparing the cfDNA characteristics to a healthy control group or a group of surgically cured NET patients, we were able to demonstrate that patients with metastatic NEN exhibit a significantly higher cfDNA concentration and percentage of *Alu* hypomethylation and a reduced *LINE-1* cfDNA integrity. These changes in cfDNA concentration, methylation, and fragmentation are strongly associated with tumor burden and can differentiate cured NET patients from patients with metastatic and advanced disease. In conclusion, our results demonstrate for the first time, that the combination of plasma cfDNA concentration, global hypomethylation, and fragmentation has the potential to serve as a potent and sensitive prognostic and therapeutic “liquid biopsy” biomarker for tumor burden and disease progression in patients with neuroendocrine neoplasias. 

## 2. Materials and Methods

### 2.1. Patient Characteristics

A group of 62 NEN patients who visited the department of endocrinology and metabolism at the University Medical Center Mainz between 2019 and 2020 was included in the study. The study was approved by the local ethics committee (vote number 2019-14664), and written informed consent was obtained from all individuals included in this study. All diagnoses are based on histopathological criteria defined by the WHO 2019. Tumor burden was evaluated by clinical and radiological evaluation, including PET-CTs, and graded semiquantitatively by the same investigator into 4 groups with no (0), low/locally defined (lymph nodes only or single distant metastases) (1), moderate (multiple distant metastases in one region, hepatic tumor load <10%) (2), or high tumor burden (multiple distant metastases in more than one region or hepatic tumor load >10%) (3). In addition, 29 healthy controls without any neoplastic disease were analyzed.

### 2.2. cfDNA Extraction and Quantification

cfDNA was extracted from 4 mL of plasma using a QIAamp^®^ MinElute^®^ ccfDNA Midi Kit (Qiagen, Hilden, Germany) according to the protocol recommended by the manufacturer. cfDNA concentration was measured using the Qubit quantification assay (Thermo Fisher Scientific, Waltham, MA, USA).

### 2.3. Estimation of cfDNA Integrity

The integrity of plasma cfDNA was evaluated by measuring long and short *LINE-1* repetitive elements fragments by qPCR (*LINE-1* 97bp- forward 5′ TGGCACATATACACCATGGAA -3′ and reverse 5′ TGAGAATGATGGTTTCCAATTTC-3′, and *LINE-1* 266bp- forward 5′ACTTGGAACCAACCCAAATG-3′ and reverse 5′CACCACAGTCCCCAGAGTG-3′) according to the protocol of Madhavan et al. [41]. For the determination of amplification efficiency, a serial dilution of commercial human genomic DNA (Promega, Madison, WI, USA) was measured for each primer set. DNA dilution was conducted with a 1:10 series, covering five dilution points (10–0.001 ng) for *LINE-1* 97-, *LINE-1* 266-, series. Concentrations of the long and short fragments were calculated by the absolute quantification method using the LightCylcer^®^ software (version 3.5.17). Thermal cycling began with an initial denaturation step (95 °C for 10 min) followed by 35 cycles of DNA denaturation (95 °C for 10 s), primer annealing (62 °C for 30 s), and primer extension (72 °C for 30 s). The temperature transition rate (C/s) was set at 20 min. cfDNA integrity was calculated according to the ratio of the concentration of long fragment to short fragment: *LINE-1* 266/97-bp.

### 2.4. Quantification of Unmethylated Alu (QUAlu) Assay 

cfDNA hypomethylation was evaluated by selective amplification of Alu repeats containing an unmethylated CpG site within the consensus sequence AACCCGG as de-scribed by Buj et al. [16]. For this purpose, in an initial step, the cfDNA was digested in parallel in two separate tubes with HpaII and MspI methylation-sensitive and -insensitive isoschizomers, respectively, generating fragments, which leave identical sticky ends (C/CGG). After ligation of a synthetic adaptor to the digested DNA fragments, the differential amplification of all amplifiable Alu elements irrespective of the methylation status and the subset of amplifiable Alu elements containing an unmethylated CpG with qPCR allowed the calculation of the fraction of unmethylated Alu elements [16]. The corresponding primer sequences for the detection of methylated *Alu* were: forward 5′AGCTACTCGGGAGGCTGAG-3′ and reverse 5′ATTCGCAAAGCTCTGACGGGTT-3′. *LINE-1* 97 was used to normalize the DNA input for both MspI and HpaII digestions. SYBR^®^Green Master Mix (Roche, Mannheim, Germany) was used for PCR reaction. Real-time PCR (qPCR) was performed with precycling heat activation of DNA polymerase at 95 °C for 10 min, followed by 40 cycles of denaturation at 95 °C for 10 s, annealing at 65 °C for 10 s. Following amplification, melting curve analysis was performed to confirm PCR product specificity and was carried out at 95 °C for 5 s, 60 °C for 60 s, and 95 °C (0.11 C/s and 5 points per C). A standard curve calculated the efficiency value for each primer pair (met. *Alu* and *LINE-1* 97-bp). Different amounts of genomic DNA ranging from 1 to 100 ng were plotted against Ct values to evaluate assay efficiency. The percentage of hypomethylated *Alu* was determined using the following formula: Percentage of unmethylated *Alu* elements =
[[(E *Alu* H)-Ct *Alu* H/(ELINE H)-Ct LINE H]/[(E *Alu* M)-Ct *Alu* M/(ELINE M)-Ct LINE M]] × 100
where E is qPCR efficiency; Ct: cycle threshold; Ct *Alu* H: Ct from qPCR using Met- *Alu* primer of HpaII digested cfDNA; Ct *Alu* M: Ct from qPCR using Met- *Alu* primer of MspI digested cfDNA; Ct *LINE-1* H: Ct from qPCR using *LINE-1* 97-bp primer of HpaII digested cfDNA; Ct *LINE-1* M: Ct from qPCR using *LINE-1* 97-bp primer of MspI digested cfDNA.

### 2.5. Statistical Analysis

The statistical analyses were carried out using Jamovi 2.0 software. Graphs and ROC analyses were performed using GraphPad Prism software (version 9.1), whereby *p* ≤ 0.05 was defined as the significance level. Since the distribution was non-normal in the Shapiro–Wilk test, non-parametric tests were applied. Data in the text and graphs are presented as median with interquartile ranges (IQR). Mann–Whitney test was applied for the comparison of patients and control groups. For comparison of patients and controls regarding tumor grade and tumor burden, Kruskal–Wallis H test was applied. If the Kruskal–Wallis test was substantial, a Dwass–Steel–Critchlow–Flinger pairwise comparison test was used for post-hoc analysis. The receiver operating characteristic (ROC) curves and multiparametric ROC curves together with the respective area under the ROC curve were calculated and used for the prediction of cutoff values and to determine the discriminatory power of cfDNA, percentage of hypomethylation, and cfDNA integrity for NEN patients. The cutoff point was calculated using the maximum value of the Youden Index (determined as sensitivity + specificity-1). Correlations of cfDNA concentration, cfDNA integrity, and hypomethylation with tumor burden and tumor grade were analyzed by Spearman Correlations. 

## 3. Results

### 3.1. Patients

The clinicopathological data of controls and patients are summarized in Table 1. The mean age of the control group was 52 years (range 24–77 years), including 12 males and 17 females. Of the 53 NEN patients with advanced metastatic disease, 28 were male, and 25 were female with an average age of 65 years. At the time of testing, the median duration of the disease was 44 months (range 2–186 months), and the median time of follow-up after the analysis was 8 months (range 3–14 months). During the follow-up period, 5 patients died. The metastatic NEN group included 7 NEC patients, 21 NET G1, 20 NET G2, and 3 NET G3 patients who all required systemic antiproliferative treatment; 14 patients had functional active tumors. In all but four patients, surgery was performed as a first-line treatment. At the time of analysis, 30 patients received somatostatin analogs, while in 5 patients, a watchful waiting strategy was followed. Additional treatments were chemotherapy in 6 patients (4 platinum/etoposide, 2 temozolomide/capecitabine), immunotherapy in 4 patients, peptide radioreceptor therapy (PRRT) in 6 patients, external radiation in 1 patient, and a combination of immunotherapy with temozolomide/capecitabine in 1 patient. Forty-four patients had primary tumors with gastroenteropancreatic origin; the others had primaries in the thymus (1), skin (1), cervix (1), unknown (1), larynx (2), and lung (3). The mean Ki67 indexes for NET G1, G2, G3, and NEC were 1.9%, 8.8%, 10.0%, and 77.5%, respectively. The group of 9 presumably cured patients all had surgically removed NET with locally confined tumor stages of well-differentiated gastrointestinal NET G1 or G2 with no clinical and no radiological signs of recurrent or metastatic disease during follow-up, which included an uneventful 68Ga-DOTATOC PET/CT in all patients. The median follow-up was 45 months (range: 18–101 months).

### 3.2. cfDNA Concentration, LINE-1 Integrity, and Alu Hypomethylation

The median plasma cfDNA concentration was significantly higher in patients with metastatic NEN (0.99 ng/µL; IQR, 0.67–1.46 ng/µL) as compared to healthy controls (0.67 ng/µL; IQR, 0.52–1.01 ng/µL; *p* = 0.004). In addition, the percentage of hypomethylated *Alu-*Gene was significantly higher in advanced NEN patients with 1.47% (IQR; 1.19–1.73%) as compared to controls with 1.23% (IQR; 1.11–1.48%, *p* = 0.031). With respect to cfDNA integrity, a significant decrease was found in NEN patients compared to healthy controls (cfDNA integrity = 0.24; IQR, 0.18–0.29 vs. 0.30; IQR, 0.23–0.33; *p* = 0.003) (Figure 1).

No differences were found between the healthy control group and the cured NET patients without any tumor manifestation, regarding plasma cfDNA concentration (0.80 ng/µL; IQR, 0.46–0.90 ng/µL), cfDNA hypomethylation (1.06% (IQR, 0.82–1.13%), and cfDNA integrity (0.290; IQR, 0.20–0.38l).

When a Spearman correlation analysis for the cfDNA characteristics in the healthy control group was performed, no association was found between cfDNA concentration, *Alu* hypomethylation, or *LINE-1* integrity (see Appendix A). Similarly, no correlation between *Alu* hypomethylation and *LINE-1* Integrity was found in the group of NEN patients (r = −0.166; *p* = 0.23). However, cfDNA concentration in NEN patients revealed a weak but significant correlation with *Alu* hypomethylation (r = 0.366; *p* = 0.01) and a moderate correlation with *LINE-1* integrity (r = −0.52; *p* = 0.001).

### 3.3. Correlation of cfDNA Characteristics with Clinical Parameters

No correlation between cfDNA concentration, *Alu* hypomethylation, or *LINE-1* integrity and age, or sex, could be found. Furthermore, in patients with advanced NEN, no correlation between cfDNA concentration, hypomethylation, or cfDNA integrity and primary location, hormonal activity of the tumor, or current treatment could be detected. Furthermore, no correlation with cfDNA characteristics and tumor differentiation (NET vs. NEC), tumor grade, or proliferation (Ki67 index) were found (for the results of these analyses, see Appendix A). When the cfDNA characteristics were correlated with the circulating Chromogranin A levels at the time of analysis, a weak but significant correlation was found for cfDNA concentration (r = 0.356; *p* = 0.011) and *Alu* hypomethylation (r = 0.310; *p* = 0.029), and a nonsignificant trend for *LINE-1* integrity (r= −0.250; *p* = 0.08), as analyzed by Spearman correlation analysis.

When the cfDNA characteristics of the groups were classified with respect to tumor burden and analyzed by Kruskal–Wallis test, the lowest cfDNA concentrations were observed in healthy controls (0.67 ng/µL; IQR 0.52–1.01 ng/µL) and in cured NET patients with no detectable tumor (0.80 ng/µL; IQR 0.46–0.90 ng/µL; not significant vs. controls), whereas patients with low or moderate tumor burden showed a concentration of 0.82 ng/µL (IQR; 0.57–1.20 ng/µL n.s. vs. control) and 1.20 ng/µL (IQR; 0.75–1.47 ng/µL; *p* = 0.040 vs. control) of cfDNA, respectively. The highest concentration of cfDNA level was found in patients with the highest tumor burden (level 3) with 3.12 ng/µL (IQR, 1.10–21.12 ng/µL; *p* < 0.0077 vs. control) (Figure 2A).

For hypomethylation, the analysis showed the lowest hypomethylation rate in healthy controls (1.23%; IQR 1.11–1.48%) and cured patients with no tumor mass (1.06%; IQR 0.82–1.13%, not significant vs. controls) (Figure 2B). When compared to healthy controls, the hypomethylation rate showed a moderate, non-significant trend for an increased hypomethylation in patients with low to moderate tumor burden (1.27%; IQR, 1.11–1.72% and 1.39%; IQR, 1.19–1.54%, respectively) while it was significantly increased in the group with the highest tumor load (4.35%; IQR, 1.78–11.09%; *p* = 0.0006). However, when the group of cured patients was compared to the patients with metastatic disease, a significant increase was found for all three tumor burden groups (Figure 2B). 

cfDNA integrity was found to be highest in healthy controls with 0.30 (IQR, 0.23–0.33), and in cured patients with 0.29 (IQR, 0.22–0.35). When compared to healthy controls, the cfDNA integrity rate showed a moderate, non-significant trend for a decrease in cfDNA integrity in patients with low (0.25; IQR, 0.23–0.30) to moderate (0.23; IQR, 0.17–0.29) tumor burden, while it was significantly decreased in the group with the highest tumor load (0.155; IQR, 0.11–0.23; *p* = 0.0085) (Figure 2C).

### 3.4. Diagnostic Power of cfDNA Characteristics for the Detection of NEN

The ROC curve analysis for plasma cfDNA concentration was able to distinguish between metastatic NEN patients and healthy individuals with an estimated AUC of 0.691 (95% CI, 0.58 to 0.80; *p* = 0.0044) with a sensitivity of 81.1% and a specificity of 41.4% at the chosen cutoff point of > 0.6 pg/µL. For *Alu* hypomethylation, the AUC was 0.645 (95% CI, 0.52–0.77; *p* = 0.0309) with a sensitivity of 58.5% and a specificity of 69% at the chosen cut off point of > 1375%, and for cfDNA integrity, the AUC was 0.697 (95% CI, 0.58−0.82; *p* = 0.0033, Figure 3A) with a sensitivity of 73.6% and a specificity of 62.1% at the chosen cutoff point of < 0.28. When ROC analysis was performed to differentiate cured NEN patients from the group of patients with advanced NEN disease a similar, not significant AUC was found for cfDNA concentration (AUC 0.674; 95% CI, 0.50 to 0.85; *p* = 0.097) and cfDNA integrity (AUC 0.666; CI, 0.47–0.86; *p* = 0.1144). However, hypomethylation revealed a very high and significant AUC of 0.874 (95% CI, 0.77–0.98; *p* = 0.004) for the presence of an advanced tumor as compared to the cured patients with the highest accuracy obtained at a cutoff > 1.135%, which corresponded to a sensitivity and specificity for the presence of an advanced NEN of 79.6% and 88.9%, respectively (Figure 3B).

When a combination of plasma cfDNA concentration, hypomethylation, and cfDNA integrity were used for the multiple logistic regression ROC analysis, the diagnostic power to discriminate patients with advanced NEN from healthy controls, the AUC was calculated with 0.694 (95% CI, 0.58 to 0.81; *p*= 0.0038) with a positive predictive value (PPV) of 70.3% and negative predictive value (NPV) of 55.6% at the chosen classification cutoff of 0.5 (Figure 4A). When the corresponding chromogranin A levels were included in the multiparametric ROC analysis, no added predictive power for the discrimination between advanced NEN from cured patients was found.

For discrimination of patients with metastasized NEN from presumably cured NEN patients, the combination of cfDNA concentration, hypomethylation, and cfDNA integrity revealed a highly significant AUC of 0.908 (95% CI, 0.83 to 0.99; *p* = 0.0001) with a positive predictive value (PPV) of 91.1% and negative predictive value (NPV) of 66.7% at the chosen classification cutoff of 0.5 (Figure 4B).

Figure 5 shows the results of the ROC analysis for the predictive power to distinguish between healthy controls or putatively cured patients and patients with advanced NEN and low, moderate, or high tumor load. While the ROC analysis showed a strong and highly significant predictive power for the prediction of advanced disease with a high tumor load as compared to healthy controls and cured NEN patients, the predictive power of most cfDNA characteristics was found to be weaker and only partially significant for the prediction of moderate and especially low tumor burden in metastasized NEN patients (for sensitivities and specificities see Appendix A). 

However, when multiparametric ROC analysis was used for cfDNA concentration, percentage of hypomethylation and cfDNA integrity to estimate the strength of the model to discriminate between healthy controls or potentially cured patients, a very strong and significant predictive power was found for all tumor burdens (Figure 6). The higher the patient’s tumor burden (TB), the better is the selectivity using ROC analysis with a negative predictive power for low, moderate, and high tumor burden of 61.8%, 73.3%, and 93.5%, and positive predictive power of 52.9%, 68.2%, and 100%, respectively (classification cutoff 0.5). For the discrimination between cured NEN patients and patients with advanced NEN disease, the discriminatory power of the multiparametric ROC analysis revealed a negative predictive power for low, moderate, and high tumor burden of 66.7%, 75%, and 75%, and positive predictive power of 86.4%, 87%, and 90.3%, respectively (classification cutoff 0.5) (for sensitivities and specificities see Appendix A).

## 4. Discussion

The concentration, integrity, and global methylation status of circulating cell-free DNA (cfDNA) is currently evaluated as a promising non-invasive blood-based tool for the diagnosis and prognosis of many cancers [9,42,43]. In contrast to tissue biopsies, liquid biopsies are faster, less invasive, have the potential to reflect all metastatic sites, can indicate therapeutic response or progression through serial sampling, and can be easily repeated over time [44]. Therefore, the purpose of our study was to assess the role of plasma cfDNA and its variables as a diagnostic and prognostic biomarker in a population of 62 patients with neuroendocrine neoplasias.

In this study, we were able to demonstrate that plasma samples from patients with advanced neuroendocrine neoplasias had significantly higher concentrations of cfDNA with a significantly stronger global *Alu* hypomethylation and reduced *LINE-1* integrity, as compared to samples from patients after curative surgery of localized NEN or a healthy control group. Furthermore, all three parameters were strongly associated with tumor burden, while no correlation was found for tumor localization, hormonal activity, or mitotic activity. *Alu* hypomethylation was the strongest predictor for the presence of a neuroendocrine tumor, both as compared to the healthy control group as well as to the surgically cured NEN patients. When cfDNA concentration, integrity, and hypomethylation status were used in combination, the predictive power was increased even further, with a positive predictive power of 86% for the detection of the presence of locally metastatic disease with low tumor burden, and increasing to 90% in patients with a moderate or high tumor burden as compared to surgically cured NEN patients. However, although significant, only a poor predictive power was found for the detection of a low NEN tumor burden as compared to a healthy control group (negative and positive predictive power of 56% and 63%, respectively). cfDNA concentration and integrity, as well as global hypomethylation, was not correlated with tumor differentiation (NET vs. NEN) and proliferation status (G1-G3) but was strongly associated with tumor load; this suggests that cfDNA characteristics represent prognostic rather than diagnostic biomarkers. Our results, therefore, support the hypothesis that cfDNA hypomethylation levels in combination with plasma cfDNA concentration and integrity are good biomarker candidates to non-invasively assess the presence of recurrent or metastatic disease, to evaluate the prognosis of the patients, and to document patient treatment response and tumor dynamics with serial measurements.

To our knowledge, this is the first study to systematically evaluate DNA integrity and methylation status of plasma cfDNA in patients with neuroendocrine neoplasias. Up to now, few studies have evaluated epigenetic modifications as prognostic markers in neuroendocrine tumor tissue. Most studies with NEN performed so far report on methylation aberration of one or a few tumor candidate genes, which usually exhibit specific promoter hypermethylation of genes associated with cell cycle, cell growth, metabolism, and DNA repair in various subgroups of neuroendocrine tumors [45,46]. However, when using plasma cfDNA from a “liquid biopsy”, DNA integrity or global hypomethylation assessed in *LINE-1*- or *Alu*-repetitive sequences could be more informative as a biomarker for the progression and prognosis of neuroendocrine tumors than the analysis of individual aberrantly methylated genes [16]. This is supported by older studies evaluating global *LINE-1* or *Alu* hypomethylation in tissue samples from neuroendocrine tumors and corresponding normal tissue. In the study of Choi et al., the tumor samples of 35 NETs were significantly less methylated than normal tissue. Hypomethylation was more prevalent in intestinal carcinoids as compared to pancreatic NET and was associated with larger tumors and lymph node metastases [18]. Similar results were reported by Stefanoli et al. where *LINE-1* hypomethylation in tissue samples from 56 pancreatic NETs was correlated with poor prognosis and advanced tumor stage [47] and by Stricker et al., who reported *LINE-1* hypomethylation in 58 GEP-NETs, which was associated with tumor grade and lymph node metastasis [48]. In some studies with GEP-NEN tumor tissue, global hypomethylation was found to be more pronounced in gastrointestinal than in pancreatic NEN [18,45]. In our study with “liquid biopsies”, however, no significant association between cfDNA plasma concentration, global hypomethylation, or DNA integrity and origin of the neuroendocrine tumor could be found. The only other study that recently evaluated the plasma cfDNA concentration in intestinal or pancreatic NETs reported higher plasma cfDNA levels in NET patients as compared to a published healthy cohort, which—in accordance to the findings of our study—was independent of primary localization, tumor grade, or proliferation. However, in contrast to our study, no further characterization of cfDNA methylation or integrity was performed, and no association with prognosis was found [49]. According to the authors, this most likely was due to the rather low tumor burden in this patient population with low-grade NET. This is supported by the results of our study, which included a significant number of advanced NEN patients with a high tumor load and accordingly showed a strong and significant association between cfDNA concentration and tumor load and very high mortality in the patient group with the highest cfDNA plasma levels, the highest hypomethylation rate, and the lowest cfDNA integrity.

Our findings of an increased concentration and hypomethylation with a decreased integrity of plasma cfDNA in neuroendocrine neoplasias are consistent with many other tumor entities. In analogy to our findings in neuroendocrine tumors, an elevated plasma cfDNA concentration and an increased hypomethylation have been postulated as a blood-based marker for advanced breast cancer, colorectal cancer, prostatic cancer, gastric cancer, or lung cancer [50,51,52,53,54].

Hypomethylation can lead to chromosomal instability, and in breast cancer, it has been postulated that *Alu* hypomethylation is a late event during cancer progression and tends to be associated with a poor prognosis [55]. In analogy, in our study, the majority of the patients who died during the observation period had very high levels of cfDNA or *Alu* hypomethylation, and all associations were strongest in the high tumor burden group, supporting the hypothesis that these markers could be of prognostic value in patients with advanced NEN. In contrast to the abundant data pointing to an important role of increased plasma cfDNA concentration and global hypomethylation in patients with metastatic neoplastic disease, the data on cfDNA integrity in cancer so far is rather sparse and, in some cases, is conflicting. However, in accordance with our results, most recent publications report on a decreased DNA integrity of *LINE-1* or *Alu* repeats in plasma cfDNA from patients with various cancers [39], such as colorectal [38,56], testicular germ cell [57], breast [16,58,59], hepatocellular [60], and ovarian cancer [61].

However, our study also has limitations, mainly related to the limited number of evaluated patients and healthy controls, especially when taking into account the heterogeneity of neuroendocrine neoplasms with regard to hormonal activity, clinical tumor behavior, prognosis, the origin of the tumors, and antiproliferative treatment modalities. In addition, although we did not find any association between *Alu* hypomethylation and *LINE-1* integrity, we did see a moderate association between cfDNA concentration and hypomethylation or integrity in advanced NEN patients. Since it would be very important to know whether cfDNA concentration and *Alu* hypomethylation or *LINE*-1 integrity are dependent or independent prognostic variables, this important question will have to be addressed in a larger study in which possible confounding variables of cfDNA concentration, integrity, and methylation status can be adequately statistically addressed. Therefore, future confirmatory multicenter studies should include a high number of patients with a broad range of tumor locations, clinical stages, and therapies. In addition, serial “liquid biopsies” with measurements of cfDNA concentration, integrity, and hypomethylation will be important in order to verify whether these markers can reflect the actual course and treatment response in NEN patients. This is particularly important as the different treatment modalities, in addition to their antiproliferative and cytoreductive effects, could also interfere directly with these markers.

Although cfDNA levels are generally higher in cancer patients than in healthy controls, it should be noted that epigenetic methylation patterns can also be influenced by environmental and lifestyle factors (like smoking) and comorbidities like chronic inflammatory diseases. To avoid confounders of the underlying disease, it would therefore be important for future studies to characterize the control group not only with respect to lifestyle factors, age, and gender, but also for inflammatory or autoimmune disease, e.g., by including additional markers such as C-reactive protein. Furthermore, traditional neuroendocrine tumor markers like chromogranin A and other novel biomarkers should also be included in a future confirmatory study and thus could potentially increase the diagnostic sensitivity in combination with cfDNA integrity and hypomethylation.

## 5. Conclusions

In conclusion, we found plasma cfDNA concentration, integrity, and hypomethylation in metastatic NEN patients to be very promising and easy to assess parameters for a “liquid biopsy” to evaluate patients with neuroendocrine neoplasms during the course of their disease. When used as a multiparameter assay, they have a very high sensitivity and specificity for the differentiation between healthy controls, cured NEN patients, and metastasized patients even with low tumor burden. Thus, the combination of plasma cfDNA concentration, integrity, and hypomethylation might be a powerful prognostic marker to facilitate therapeutic decisions and monitor the therapy and the course of the disease in patients with advanced NEN. 

## Figures and Tables

**Figure 1 cancers-14-01025-f001:**
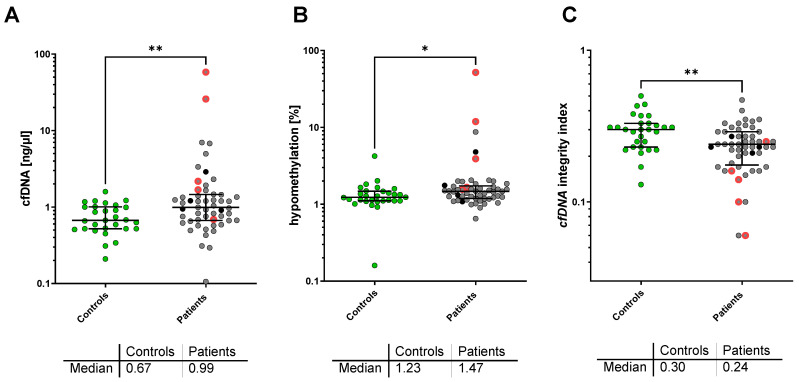
Plasma cfDNA concentration (**A**), cfDNA *Alu* hypomethylation (**B**), and *LINE-1* cfDNA integrity (**C**) in the healthy control group (green) and patients with metastatic NEN (grey). Deceased patients are marked in red, patients lost to follow-up in black. * *p* < 0.05; ** *p* < 0.005 in the Mann–Whitney U-test.

**Figure 2 cancers-14-01025-f002:**
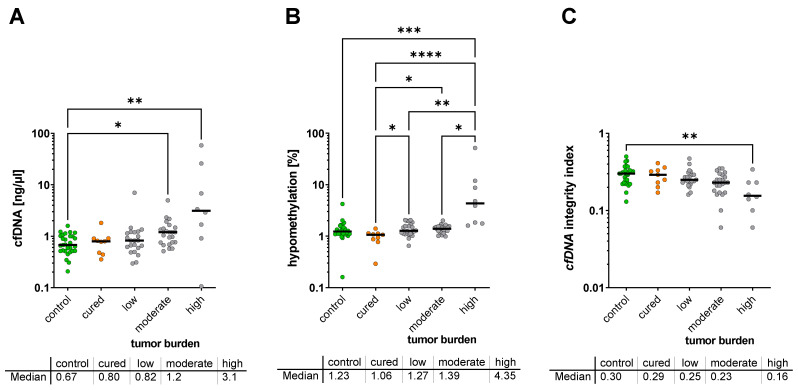
Plasma cfDNA concentration (**A**), cfDNA hypomethylation (**B**), and cfDNA integrity (**C**) with respect to tumor burden (no tumor, low, moderate, or high tumor burden) * *p* < 0.05.; ** *p* < 0.01; *** *p* < 0.001; **** *p* < 0.0001.

**Figure 3 cancers-14-01025-f003:**
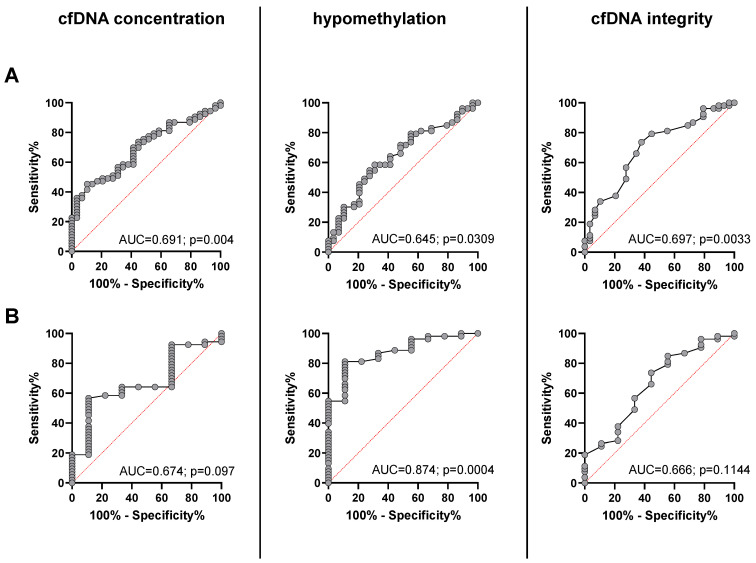
Receiver operating characteristic (ROC) curves to distinguish between healthy controls and patients with advanced NEN disease (**A**) and between presumably cured NEN patients and patients with advanced NEN (**B**) for cfDNA concentration (**left**), percentage of cfDNA hypomethylation (**middle**), and cfDNA integrity (**right**).

**Figure 4 cancers-14-01025-f004:**
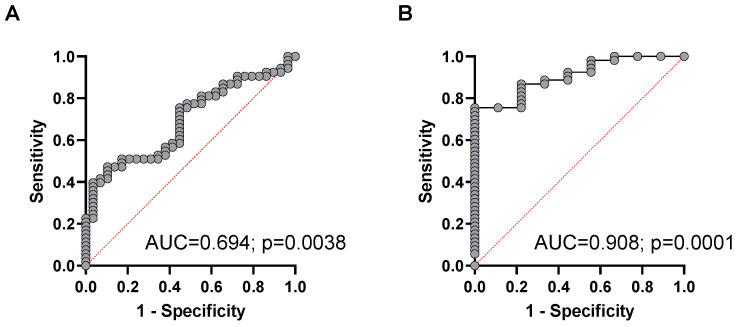
Multiparametric ROC analysis using plasma cfDNA concentration, percentage of cfDNA hypomethylation, and cfDNA integrity to estimate the strength of the model to discriminate between (**A**) healthy controls and NEN patients with advanced disease or between (**B**) presumably cured NEN patients and patients with advanced NEN disease.

**Figure 5 cancers-14-01025-f005:**
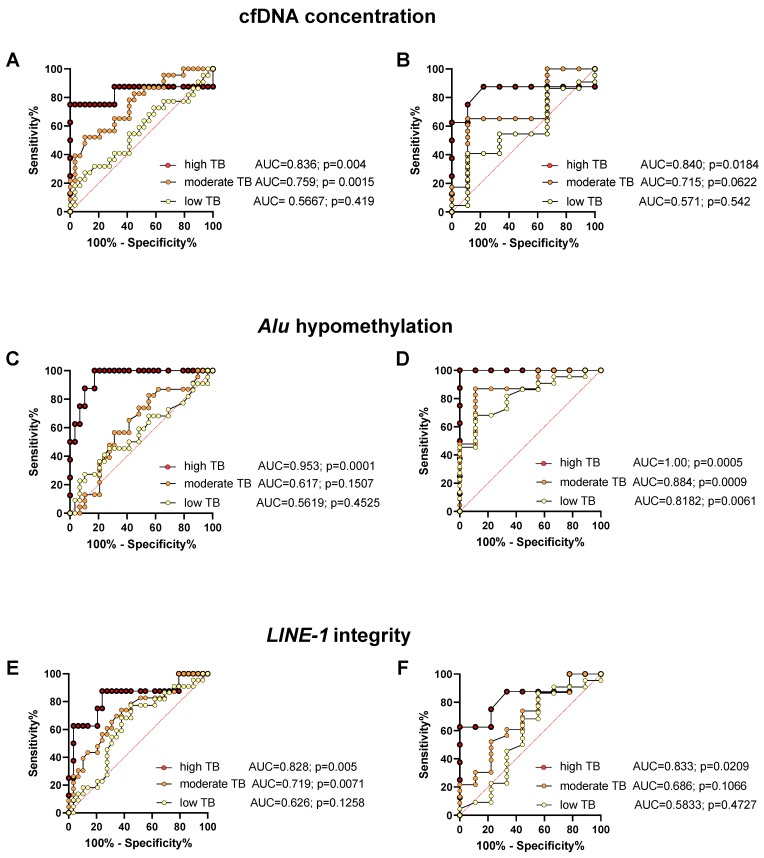
ROC analysis of plasma cfDNA concentration (**A**,**B**), *Alu* hypomethylation (**C**,**D**), and *LINE-1* integrity (**E**,**F**) for distinguishing healthy controls (**A**,**C**,**E**), or cured patients (**B**,**D**,**F**) from patients with low, moderate, and high tumor burden (TB), along with the area under the curve (AUC), for sensitivities and specificities see Appendix A.

**Figure 6 cancers-14-01025-f006:**
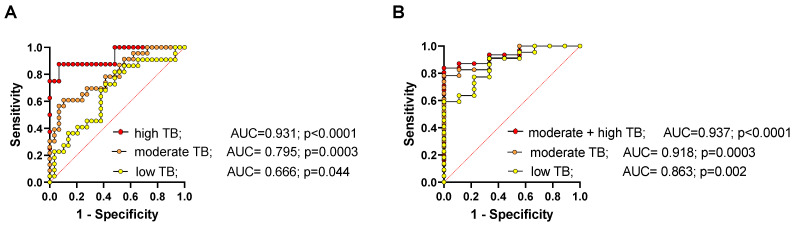
Multiparametric ROC analysis using cfDNA concentration, percentage of *Alu* hypomethylation, and *LINE-1* integrity to estimate the strength of the model to discriminate between (**A**) healthy controls and NEN patients or (**B**) presumably cured NEN patients and patients with low, moderate, and high tumor burden (TB), along with the area under the curve (AUC), for sensitivities and specificities see Appendix A.

**Table 1 cancers-14-01025-t001:** Clinicopathological features of the study groups. Detailed baseline clinicopathological features of the study groups, including sex, age, tumor differentiation grade (G1: Grade 1, G2: Grade 2, G3: Grade 3, N/A-not available), and tumor burden (low, moderate, and high).

Groups/ Subgroups	Characteristics											
	N	Age	Gender		Grading				Tumor Burden at Time of Analysis			
		Mean (Range)	Men	Woman	G1	G2	G3	N/A	0	Low	Mod	High
Controls	29	52 (24–77)	12	17	-	-	-	-	29	-	-	-
NEN Patients	53	65 (33–87)	28	25	21	20	10	2	-	22	23	8
NET	46	65 (33–87)	25	21	21	20	3	2	-	21	20	5
NEC	7	61 (41–82)	3	4	0	0	7	-	-	1	3	3
Cured NET patients	9	62 (33–75)	5	4	7	2	-	-	9	-	-	-

## Data Availability

The data presented in this study are available on request from the corresponding author.

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
