# Peer review of "Quantitative Analysis of Plasma Cell-Free DNA and Its DNA Integrity and Hypomethylation Status as Biomarkers for Tumor Burden and Disease Progression in Patients with Metastatic Neuroendocrine Neoplasias"

_cancers, 2022, doi:10.3390/cancers14041025_

Round 1
Reviewer 1 Report
The authors found plasma cfDNA concentration, integrity, and hypomethylation in metastatic NEN patients to be very promising and easy to assess parameters for a “liquid biopsy” to evaluate patients with neuroendocrine neoplasms during the course of their disease. Overall, the key idea is novel, the experiments are well designed, the results are solid and sufficient, the manuscript is well organized.
I am just wondering whether calculating the auPRC score is better than the auROC score? Because for prognostic and therapeutic biomarker for tumor burden and disease progression, it is more important to measure the precision and recall. The authors can try to evaluate the results using auPRC scores.
Author Response
We would like to thank all reviewers for taking the time to review our manuscript and for their valuable comments. We have incorporated all suggestions into the revised manuscript and have the feeling, that due to the recommended changes the manuscript has gained in strength. Enclosed please find the response to the comments of Reviewer 1.
Review Report 1
I am just wondering whether calculating the auPRC score is better than the auROC score? Because for prognostic and therapeutic biomarker for tumor burden and disease progression, it is more important to measure the precision and recall. The authors can try to evaluate the results using auPRC scores
The suggestion of the reviewer to calculate the novel auPRC score and compare it to the auROC is very interesting, since the auPRC score has indeed recently been discussed to be a promising new tool for risk prediction. In our paper, we use the receiver operating characteristic (ROC) curve since in medical research the auROC score currently is the most popular tool for evaluating the prediction accuracy of a risk model and probably will best be understood by the majority of the readers. However, for future studies on the role of cfDNA integrity and methylation in NEN, it indeed would be very interesting to compare the predictive power of both tools.
Reviewer 2 Report
- It would be interesting to kwon if the findings described (cfDNA, hypometilation and DNA intregity) are correlated wiht the Chromogranin A levels.
- Section 2.1(patiens characteristics) should describe teh differents treatments in the patiens with advance disease
3. It would be important to comment in the discussion if the differents treatments (somatostain analogs, everolimus, sunitinib..) can interfer with the results.
Author Response
We would like to thank all reviewers for taking the time to review our manuscript and for their valuable comments. We have incorporated all suggestions into the revised manuscript and have the feeling, that due to the recommended changes the manuscript has gained in strength. Enclosed please find the response to the comments of Reviewer 2.
Review Report 2
- It would be interesting to know if the findings described (cfDNA, hypometilation and DNA intregity) are correlated with the Chromogranin A levels.
This indeed is a very interesting and important question. We have performed the requested analysis. Chromgranin A showed a weak but significant correlation with cfDNA concentration and Alu hypomethylation, but did not add any predictive power when combined with the parameters of the cfDNA in the AUC analysis comparing cured versus advanced NEN patients. These results are now stated in the result section of the manuscript.
- Section 2.1(patients characteristics) should describe the different treatments in the patients with advanced disease. It would be important to comment in the discussion if the different treatments (somatostain analogs, everolimus, sunitinib..) can interfer with the results.
Thank you very much for bringing up this important issue. The information on the treatment is now given in the section on patient characteristics and results and possible interaction of the results is now discussed in the paper. Correlation analysis of treatment and cfDNA properties has been added in the supplement (Table S2). In our crossectional study, we did not find any significant correlation of the different treatment groups. Therefore, a possible influence of the different treatment modalities will be evaluated in a future longitudinal study, which is planned as a follow-up study in our center and which will include patients before, during, and after treatment.
Reviewer 3 Report
In the present work, Mettler, Fottner and colleagues describe the detection of cell-free circulating tumor DNA (cfDNA) concentration, integrity and methylation as possible non-invasive methods for the diagnostic evaluation and stratification of patients with neuroendocrine tumors.
The work is introduced by a description of the pathological context and a state of the art of available biomarkers for the diagnostic/prognostic evaluation of patients affected by these neoplasia. The Authors then describe different strategies of detection of the plasma tumor DNA and provide a description of advantages and limitations of these markers. They also describe the relevance of DNA methylation in tumor biology, by detailing the current knowledge on neuroendocrine neoplasia (NEN). They therefore propose that cell-free circulating tumor DNA amount, integrity and methylation status might be informative on tumor burden and clinical evolution in these patients. As for many other cancer types, the identification of novel non-invasive tools for rapid diagnosis and prognosis is of considerable importance.
Overall, the article is well written, even if some sparce typos make some statements unclear (I detailed some as minor points below). The introduction provide sufficient elements for a non-specialist to understand the relevance of the questions, the methodological advantages and limitations, and overall allows the reader to understand the narrative and the strategy. In particular, the Authors detail LINE-1 regions as a surrogate for the evaluation of global DNA methylation in cancer cells. The study seems sufficiently well designed, and includes clinical data extracted from the follow-up of patients considered as cured or with variable tumor burden and different clinical outcomes (such as metastatic evolution). The study also includes samples obtained from 29 control individuals with no clinical signs of NEN.
However, I have several concerns on the data presented and on the relevance of the conclusions made by the Authors.
- The Authors make multiple conclusions based on data that are not presented. I understand that space limitations do not allow to show all the comparisons that do not reach statistical significance, but all the data described should be clearly presented as supplementary information to support the conclusions.
- The results presented figure 1 display very moderate but statistically significant differences. However, there is a very large overlap between the values measured in control and metastatic patients, and differences seem to be driven especially by the values measured in some of the patients deceased during the follow-up. Can the Authors identify those patients (for example providing tags and results in a supplementary table) to examine whether the 3 datasets correlate? Is there any correlation between the integrity of the LINE-1 sequences and the quantification of the methylation status? Can the author convincingly prove that the measure of LINE-1 hypomethylation (which is based on quantitative amplification of the template regions after digestion with methylation-sensitive restriction enzymes) is not biased by the variable integrity of the LINE-1 template itself?
- It would be interesting to know whether methylation of LINE-1 at specific genomic locations would be more informative than the global results. However, mapping repetitive sequences would require sophisticated -omic analyses. If methylome data are not available, it is unreasonable to ask. However, have the Authors attempted to examine the methylation of different elements, like Intracisternal-A particles (IAP) to see whether hypomethylation is generalized or restricted to defined genomic regions?
- At line 254, the Authors start describing putative correlations with cfDNA concentration, integrity and methylation, highlighting a significant correlation between these 3 parameters and tumor burden. However, the tumor burden is represented as a grading in different classes. Where are the data? It is unacceptable, in my opinion, to perform a statistical method of correlation between a quantitative continuous measure (DNA concentration, methylation and integrity) and a qualitative (i.e. somehow arbitrary) classification. Unless the Authors can provide a robust quantitative measure of tumor burden, these correlations are purely speculative and the reader must stick to the comparison presented in panels 2A, 2B and 2C.
- When the Authors describe the diagnostic power of cfDNA-associated measures, some key information are systematically omitted (even if some extrapolation is possible by looking at the curves). The ROC analysis should serve as method to describe the correlation between sensitivity and specificity, therefore ultimately defining a value representing the best compromise. In particular, I think many would agree that specificity is of great importance since it defines the ability of diagnostic/prognostic tests to avoid false positives, e.g. non-metastatic patients that the test would predict as at risk of metastatic progression. Indeed, Authors should either present these information as a table associated with the respective panel, or they should clearly state examples of threshold values with their associated sensitivity and specificity for all ROC analyses, in order the reader can evaluate the performance of the examined characteristics.
Minor points:
- The end of the statement starting at line 16 is unclear. A “be” is probably missing after “could” in the sentence at line 18.
- There is a repetition in the sentence starting at line 67: “hampered by low specificity, variable sensitivity, and poor specificity…”.
- Line 181: primer annealing for the 30s
- At line 385, the Authors mentioned that “few studies have investigated the epigenetic modifications DNA methylation in NEN”. I think the present study does not add any evidence on the functional biological role of DNA methylation, but attempts to correlate one parameter (DNA methylation) with clinical outcome. Indeed, they should not use the present lack of knowledge to overstate the advance represented by the work.
- Minor spell check is required.
Author Response
We would like to thank all reviewers for taking the time to review our manuscript and for their valuable comments. We have incorporated all suggestions into the revised manuscript and have the feeling, that due to the recommended changes the manuscript has gained in strength. Enclosed please find the response to the comments of Reviewer 3.
Review Report 3
- The Authors make multiple conclusions based on data that are not presented. I understand that space limitations do not allow to show all the comparisons that do not reach statistical significance, but all the data described should be clearly presented as supplementary information to support the conclusions.
As suggested by the reviewer all data described, including the comparisons that did not reach statistical significance, are now given in a supplemental section (table S1, or in the main paper).
- The results presented figure 1 display very moderate but statistically significant differences. However, there is a very large overlap between the values measured in control and metastatic patients, and differences seem to be driven especially by the values measured in some of the patients deceased during the follow-up. Can the Authors identify those patients (for example providing tags and results in a supplementary table) to examine whether the 3 datasets correlate?
As suggested by the reviewer, the deceased patients and the outliers are now numbered individually and given in a supplementary figure S1.
- Is there any correlation between the integrity of the LINE-1 sequences and the quantification of the methylation status? Can the author convincingly prove that the measure of LINE-1 hypomethylation (which is based on quantitative amplification of the template regions after digestion with methylation-sensitive restriction enzymes) is not biased by the variable integrity of the LINE-1 template itself?
We are thankful to the reviewer for bringing up this important question of a possible interaction between the different variables.
From a methodological point of view, we can rule out the possibility that the results of the methylation status are confounded by the LINE-1 integrity, since we measured the hypomethylation status in the Alu sequences and not in the LINE-1 sequences, while DNA integrity was evaluated in LINE-1. In the method section, it is described that for the analysis of Alu-hypomethylation a short LINE-1 97bp template was used. However, this was utilized only as an internal control DNA fragment to normalize the DNA concentration between the two methylation samples within the same probe of one patient.
From a statistical point of view, we performed a correlation analysis, which showed no associations within the control group. Similarly, in NEN patients no correlation between LINE-1-integrity and Alu-hypomethylation was seen. However, for cfDNA concentration, we found a weak but statistically significant correlation between LINE-1 integrity or Alu-hypomethylation in the NEN patient group. The implications of this, however, remain unclear, since this cannot inform us on whether these associations are due to a causal relationship. Since from a clinical point of view it would of course be very important to know whether cfDNA concentration and Alu-hypomethylation or LINE-1 integrity are dependent or independent prognostic variables, this important question will have to be addressed in a future larger study, in which possible confounding variables of cfDNA concentration, integrity, and methylation status can be included and statistically adequately addressed.
This important issue is now discussed in the paper, and the data of the correlation analysis between the variables is now given in the result section.
- It would be interesting to know whether methylation of LINE-1 at specific genomic locations would be more informative than the global results. However, mapping repetitive sequences would require sophisticated -omic analyses. If methylome data are not available, it is unreasonable to ask. However, have the Authors attempted to examine the methylation of different elements, like Intracisternal-A particles (IAP) to see whether hypomethylation is generalized or restricted to defined genomic regions?
This is an interesting question, however, methylome data is not available and we have not examined the methylation status of other elements yet.
- At line 254, the Authors start describing putative correlations with cfDNA concentration, integrity and methylation, highlighting a significant correlation between these 3 parameters and tumor burden. However, the tumor burden is represented as a grading in different classes. Where are the data? It is unacceptable, in my opinion, to perform a statistical method of correlation between a quantitative continuous measure (DNA concentration, methylation and integrity) and a qualitative (i.e. somehow arbitrary) classification. Unless the Authors can provide a robust quantitative measure of tumor burden, these correlations are purely speculative and the reader must stick to the comparison presented in panels 2A, 2B and 2C.
As correctly stated by the reviewer, a semiquantitative measure has been used for the evaluation of tumor burden. Although we believe, that in principle a correlation between a quantitative continuous measure and a qualitative classification can statistically be performed, we agree with the reviewer that – given the semiquantitative assessment of tumor burden - it is advisable to rely mainly on the comparisons presented in panel 2. Therefore, the correlation data in this section are replaced by a more descriptive approach as suggested by the reviewer.
- When the Authors describe the diagnostic power of cfDNA-associated measures, some key information are systematically omitted (even if some extrapolation is possible by looking at the curves). The ROC analysis should serve as method to describe the correlation between sensitivity and specificity, therefore ultimately defining a value representing the best compromise. In particular, I think many would agree that specificity is of great importance since it defines the ability of diagnostic/prognostic tests to avoid false positives, e.g. non-metastatic patients that the test would predict as at risk of metastatic progression. Indeed, Authors should either present these information as a table associated with the respective panel, or they should clearly state examples of threshold values with their associated sensitivity and specificity for all ROC analyses, in order the reader can evaluate the performance of the examined characteristics.
As suggested by the reviewer threshold values with their associated sensitivity and specificity for all ROC analyses are now stated for all ROC analyses either in the main manuscript or as a table in the supplement section (table S2).
Minor points:
- The end of the statement starting at line 16 is unclear. A “be” is probably missing after “could” in the sentence at line 18.
- There is a repetition in the sentence starting at line 67: “hampered by low specificity, variable sensitivity, and poor specificity…”.
- Line 181: primer annealing for the 30s
These points were corrected accordingly in the article.
- At line 385, the Authors mentioned that “few studies have investigated the epigenetic modifications DNA methylation in NEN”. I think the present study does not add any evidence on the functional biological role of DNA methylation, but attempts to correlate one parameter (DNA methylation) with clinical outcome. Indeed, they should not use the present lack of knowledge to overstate the advance represented by the work.
According to the suggestion of the reviewer, the limitations of the current study are now mentioned more clearly in the discussion to avoid overstatements.
- Minor spell check is required.
A spell-check of the revised paper was performed
Reviewer 4 Report
The authors provide an interesting finding about plasma circulating cell free DNA (cfDNA) as diagnostic prognostic marker for neuroendocrine neoplasia, which has relatively few biomarker. They show that cfDNA from NENs have higher cfDNA and lower methylation and less integrity in the cfDNA as determined by the LINE1 elements. Some comments are :
The authors can provide a little bit details about tumor burden and how the categories are devised. Could the authors provide more combinations about ROC analysis with respect to tumor burden? It will beneficial to find good AUC for low to medium categories. How well is other combinations for eg cfDNA and hypomethylation or cfDNA total and integrity work? Is there any difference between normal controls with presumably cured? The authors can clarify presumably cured as well i.e how was it defined from a clinical point of view.
“No correlation between cfDNA concentration, Alu hypomethylation, or Line1-integreity and age, or sex, could be found.” Was this within the same group? Did any control sample show any significance with respect to cfDNA conc or integrity? Can the authors clarify which primers are used from reference 41?
Author Response
We would like to thank all reviewers for taking the time to review our manuscript and for their valuable comments. We have incorporated all suggestions into the revised manuscript and have the feeling, that due to the recommended changes the manuscript has gained in strength. Enclosed please find the response to the comments of Reviewer 4.
Review Report 4
- The authors can provide a little bit details about tumor burden and how the categories are devised.
In accordance with the ethical committee approval, no additional radiological evaluation was performed for this study. Therefore, tumor burden at the time of the cfDNA analysis was assessed semiquantitatively based on the latest radiological scans and PET-CT analysis. The tumor load was graded into 4 groups, no (0), low/locally defined (lymph nodes only or single distant metastases) (1), moderate (multiple distant metastases in one region, hepatic tumor load < 10%) (2), or high tumor burden (multiple distant metastases in more than one region or hepatic tumor load > 10%) (3). According to the suggestion of the reviewer, this is now outlined in the method section more clearly.
- Could the authors provide more combinations about ROC analysis with respect to tumor burden? It will beneficial to find good AUC for low to medium categories. How well is other combinations for eg cfDNA and hypomethylation or cfDNA total and integrity work?
We absolutely agree with the reviewer, that it would be of great clinical relevance to have high predictive power in the ROC analysis for the discrimination between healthy controls or putatively cured patients and patients with metastasized NEN, especially with low and moderate tumor load. In accordance with the suggestion of the reviewer, we have therefore performed multiparametric analyses with all possible combinations including the combination of only two of these cfDNA parameters.
As depicted in Fig 4 of the manuscript, we found the best results when a combination of plasma cfDNA concentration, Alu-hypomethylation, and LINE-1 integrity was used for the multiple logistic regression ROC analysis. However, the results of the other multiparametric ROC analyses are now given in table S3 in the supplement section of the manuscript.
- Is there any difference between normal controls with presumably cured? The authors can clarify presumably cured as well i.e how was it defined from a clinical point of view.
As depicted in figure 2 of the manuscript, no significant differences between the normal controls and the presumably cured patients were detected with respect to cfDNA concentration, Alu-hypomethylation, and LINE-1 integrity. Furthermore, no significant differences with respect to sex and age were found between the cured patients, the control group, or the advanced patients. However, it has to be kept in mind that the number of cured patients was only small.
The group of 9 presumably cured patients all had surgically removed NET with locally confined tumor stages of well-differentiated gastrointestinal NET G1 or G2 with no clinical and no radiological signs of recurrent or metastatic disease during follow-up which included an uneventful 68Ga-DOTATOC PET/CT in all patients. The median follow-up was 45 months (range 18-101 months). As suggested by the reviewer, this information is now given in greater detail in the manuscript.
- “No correlation between cfDNA concentration, Alu hypomethylation, or Line1-integreity and age, or sex, could be found.” Was this within the same group? Did any control sample show any significance with respect to cfDNA conc or integrity?
This is an important question. Indeed no correlation between age, sex, and cfDNA characteristics could be detected both within the control group as well as within the patient group. The data is now shown in the supplement Fig. S1.
- Can the authors clarify which primers are used from reference 41?
As detailed in the method section, the integrity of plasma cfDNA was evaluated by measuring long and short LINE-1 repetitive elements fragments by qPCR (LINE-1 97bp- forward 5´ TGGCACATATACACCATGGAA -3´ and reverse 5´ TGAGAATGATGGTTTCCAATTTC-3´, and LINE-1 266bp- forward 5´ACTTGGAACCAACCCAAATG-3´ and reverse 5´CACCACAGTCCCCAGAGTG-3´) according to the protocol of Madhavan et al [41].
This question might have come up due to an incorrect citation numbering in the manuscript version of the reviewer portal. However, this formatting problem has now been corrected, so that the citations are again in accordance with the original manuscript.
Round 2
Reviewer 3 Report
Authors have appropriately answered most of the points raised by myself and other Reviewers. The manuscript has been improved and the work is overall well presented and clear.
Before the manuscript is accepted for publication, I would like to stress once again the importance of a possible correlation between the Alu methylation status and the integrity of DNA. DNA methylation of Alu sequences between two biological sources (i.d. control patients vs tumor patients or cured patients vs metastatic patients) can only be compared using such an amplification-related methodology if the integrity of the template across samples is identical. Given the fact that circulating LINE-1 sequence integrity is altered in tumor patients, I am not convinced that Authors can assume that Alu integrity is unaffected without an instrumental proof. This should be done, for example by comparing the integrity (as size) of template Alu sequences or amplicon bands from undigested samples from control and tumor samples to prove identical profiles.
Author Response
Authors have appropriately answered most of the points raised by myself and other Reviewers. The manuscript has been improved and the work is overall well presented and clear.
Before the manuscript is accepted for publication, I would like to stress once again the importance of a possible correlation between the Alu methylation status and the integrity of DNA. DNA methylation of Alu sequences between two biological sources (i.d. control patients vs tumor patients or cured patients vs metastatic patients) can only be compared using such an amplification-related methodology if the integrity of the template across samples is identical. Given the fact that circulating LINE-1 sequence integrity is altered in tumor patients, I am not convinced that Authors can assume that Alu integrity is unaffected without an instrumental proof. This should be done, for example by comparing the integrity (as size) of template Alu sequences or amplicon bands from undigested samples from control and tumor samples to prove identical profiles.
Thank you for bringing up this important issue again. We now can see the point of the reviewer more clearly. Indeed, our data does not allow to conclude, that Alu-integrity is not affected in patients with NEN, in which we have demonstrated an altered LINE-integrity.
However, even in the presence of alterations of Alu-integrity this should not have a major impact on the results of Alu-hypomethylation, as measured with the method of our paper. As now described in greater detail in the method section, the principle of the assay used for quantification of unmethylated Alu (QUAlu) in our study is the selective amplification of short Alu repeats containing an unmethylated CpG site within the consensus sequence AACCCGG. For this purpose, in an initial step the cfDNA is digested in parallel in two separate tubes with HpaII and MspI methylation-sensitive and -insensitive isoschizomers, respectively, generating fragments which leave identical sticky ends (C/CGG). After ligation of a synthetic adaptor to the digested DNA fragments the differential amplification of all amplifiable Alu elements irrespective of the methylation status and the subset of amplifiable Alu elements containing an unmethylated CpG with qPCR allows the calculation of the fraction of unmethylated Alu elements, which has been reported to be altered in a variety of tumors and is believed to be a surrogate parameter of global hypomethylation (Ref. 16).
Therefore, since for evaluation of the Alu-hypomethylation status the target cfDNA is enzymatically digested as an initial step, the integrity of the starting cfDNA should not significantly affect the results of the hypomethylation assay, used in our publication and by many other groups in different tumor entities. Indeed, the robustness of the QUAlu assay with respect to the integrity of the starting DNA has been already demonstrated convincingly in the methodology paper by Buj R et al. 2016 (Ref. 16). In this paper it was demonstrated that the assay is relatively unaffected by the quantity and quality of the starting material and even shows comparable results between high quality DNA and artificially degraded DNA aliquots sheared by enzymatic digestion or sonication (for details see Ref 16, supplementary figure S3 B/C).
In order to present these methodological aspects more clearly to the reader, we now describe the principle of the method in the paper in greater detail, and hope that we were able to address the important concerns of the reviewer adequately.